# Landscape of Preterm Birth Therapeutics and a Path Forward

**DOI:** 10.3390/jcm10132912

**Published:** 2021-06-29

**Authors:** Brahm Seymour Coler, Oksana Shynlova, Adam Boros-Rausch, Stephen Lye, Stephen McCartney, Kelycia B. Leimert, Wendy Xu, Sylvain Chemtob, David Olson, Miranda Li, Emily Huebner, Anna Curtin, Alisa Kachikis, Leah Savitsky, Jonathan W. Paul, Roger Smith, Kristina M. Adams Waldorf

**Affiliations:** 1Department of Obstetrics and Gynecology, University of Washington, Seattle, WA 98195, USA; brahm.coler@wsu.edu (B.S.C.); smccart@uw.edu (S.M.); miranda.li2@columbia.edu (M.L.); huebnem@uw.edu (E.H.); anna14@uw.edu (A.C.); abk26@uw.edu (A.K.); savitsky@uw.edu (L.S.); 2Elson S. Floyd College of Medicine, Washington State University, Spokane, WA 99202, USA; 3Department of Physiology, University of Toronto, Toronto, ON M5S 1A8, Canada; shynlova@lunenfield.ca (O.S.); aboros@lunenfield.ca (A.B.-R.); lye@lunenfield.ca (S.L.); 4Department of Obstetrics and Gynecology, University of Toronto, Toronto, ON M5G 1E2, Canada; 5Department of Obstetrics and Gynecology, University of Alberta, Edmonton, AB T6G 2R7, Canada; kelycia@ualberta.ca (K.B.L.); wendy2@ualberta.ca (W.X.); dmolson@ualberta.ca (D.O.); 6Departments of Pediatrics, Université de Montréal, Montréal, QC H3T 1J4, Canada; sylvain.chemtob@gmail.com; 7Departments of Pediatrics and Physiology, University of Alberta, Edmonton, AB T6G 2S2, Canada; 8Department of Biological Sciencies, Columbia University, New York, NY 10027, USA; 9Mothers and Babies Research Centre, School of Medicine and Public Health, College of Health, Medicine and Wellbeing, University of Newcastle, Callaghan, NSW 2308, Australia; jonathan.paul@newcastle.edu.au (J.W.P.); roger.smith@newcastle.edu.au (R.S.); 10Hunter Medical Research Institute, 1 Kookaburra Circuit, New Lambton Heights, NSW 2305, Australia; 11John Hunter Hospital, New Lambton Heights, NSW 2305, Australia; 12Department of Global Health, University of Washington, Seattle, WA 98195, USA

**Keywords:** tocolytic, preterm birth, preterm labor, neonate, prematurity, pregnancy, therapeutic, progesterone, fetus

## Abstract

Preterm birth (PTB) remains the leading cause of infant morbidity and mortality. Despite 50 years of research, therapeutic options are limited and many lack clear efficacy. Tocolytic agents are drugs that briefly delay PTB, typically to allow antenatal corticosteroid administration for accelerating fetal lung maturity or to transfer patients to high-level care facilities. Globally, there is an unmet need for better tocolytic agents, particularly in low- and middle-income countries. Although most tocolytics, such as betamimetics and indomethacin, suppress downstream mediators of the parturition pathway, newer therapeutics are being designed to selectively target inflammatory checkpoints with the goal of providing broader and more effective tocolysis. However, the relatively small market for new PTB therapeutics and formidable regulatory hurdles have led to minimal pharmaceutical interest and a stagnant drug pipeline. In this review, we present the current landscape of PTB therapeutics, assessing the history of drug development, mechanisms of action, adverse effects, and the updated literature on drug efficacy. We also review the regulatory hurdles and other obstacles impairing novel tocolytic development. Ultimately, we present possible steps to expedite drug development and meet the growing need for effective preterm birth therapeutics.

## 1. Introduction

Though preterm birth (PTB) is the leading cause of infant morbidity and mortality, it lacks clear therapeutic options. Approximately 15 million preterm neonates are born annually and more than 1 million die from subsequent complications within the following 5 years [1,2]. Adverse outcomes related to prematurity span a spectrum that includes neurodevelopmental (e.g., neurocognitive impairment, cerebral palsy, hearing impairments, and intellectual or motor disabilities), pulmonary (e.g., bronchopulmonary dysplasia) and ophthalmologic (e.g., retinopathy of immaturity) disorders that contribute to lifelong disability [3,4,5]. The earlier the gestational age at PTB, the greater the risk for prematurity-related adverse outcomes [6]. Thus, PTB presents a formidable health risk to the neonate and a compelling condition for which therapeutics could lessen the burden of disease and disability across a child’s lifespan.

Tocolytic agents are drugs used to delay spontaneous preterm labor with intact membranes and comprise the focus of this review. We refer to tocolytic agents and select other drugs targeting other PTB pathways as ‘preterm birth therapeutics’. Typically, these agents delay delivery only for a short time (i.e., 48 h or less; Table 1). The pathophysiology underlying PTB is multi-faceted yet largely predicated on premature activation of parturition pathways within the myometrium or placenta; many therapeutics are, therefore, tailored towards the specific etiologies of PTB. Historically, the approaches for reducing uterine contractions to delay PTB have followed two main pathways: (1) decreasing contractile proteins in the myometrium or (2) inhibiting the synthesis of myometrial stimulants [7]. A short delay prior to delivery can be crucial to provide an opportunity to administer antenatal corticosteroids to accelerate fetal lung development or to transfer the patient to a facility with neonatal intensive care [7]. Interestingly, most tocolytics have not been shown to be effective in improving neonatal outcomes, despite the efficacy in delaying PTB [8]. Combination therapies that involve tocolytic drugs and differing mechanisms of action are being studied, as it has been posited that combination therapy can not only improve the overall tocolytic impact but also reduce dosage, frequency of administration, and adverse effects [7]. Preliminary results from some studies, however, suggest no clear advantage for combination therapy over individual therapeutic administration [7]. Most tocolytics have been shown to be ineffective in further delaying PTB when used as maintenance therapy [9,10,11,12,13]. The choice for first-line tocolytic therapy is still controversial, as each class varies in effectiveness, safety profile, cost, and availability.

Globally, there is a significant need for better tocolytic agents to prevent PTB. The global PTB rate has risen steadily from 9.8% in 2000 to 10.6% in 2014 and ranges from 5–18%, depending on country and race/ethnicity [14,15]. The vast majority of PTBs occur in lower- and middle-income countries (LMICs)—particularly in southern Asia and sub-Saharan Africa [15]. LMICs in these regions account for 52% of global livebirths but more than 80% of PTBs [2,15]. In high-income countries, a neonate born very preterm (28–32 weeks) has a 95% chance of survival compared to only a 30% chance for neonates born in many low-income countries [16]. There are notable outliers, however, as some lower-income countries, such as Ecuador, have a low PTB rate of 5% [15]. Neighboring countries Uganda and Tanzania, of relatively similar economic standing and healthcare infrastructure, have strikingly different rates at 6.6% and 16.6%, respectively [15]. The United States is a high-income country and is another outlier, as it is included among the top ten countries with the highest numbers of PTBs [2].

Although the unmet need for new PTB therapeutics is significant, a combination of a relatively small market size and formidable regulatory hurdles has led to a stagnant drug pipeline with minimal pharmaceutical interest. Recently, discoveries using new therapeutic strategies and insights into the mechanism of parturition have increased the potential for new drugs with enhanced efficacy to be developed and approved for PTB. These discoveries have spurred the few ongoing pre-clinical and clinical trials currently studying the efficacy and safety profiles of novel therapeutics (Table 2). In this review, we focus on the dynamic landscape of PTB therapeutics, including drugs that have been used historically as well as new ones in development: ritodrine, terbutaline, atosiban, nifedipine, antibiotics, broad spectrum chemokine inhibitors (BSCIs), interleukin-1 (IL-1R) receptor antagonists, liposomes, and nanoparticle platforms. Agents administered to address more complex diseases associated with PTB (e.g., fetal growth restriction or emergent hypertensive disorders) are not discussed in this review. Discussion of mechanical therapies to prevent PTB, such as the cerclage and the cervical pessary, are outside the scope of this review [17,18,19]. When possible, we present the history of drug development, mechanism of action, side effects, and updated literature on drug efficacy. We also review the numerous regulatory hurdles and other obstacles for tocolytics in development and propose a path forward to expedite therapeutic advancements for the benefit of pregnant people and their neonates.

## 2. Pathophysiology of PTB

The pathophysiology of PTB is complex, brought on by a myriad of pathologic processes or underlying conditions that include activation of the fetal hypothalamic-pituitary-adrenal axis, infection and inflammation, decidual hemorrhage or thrombosis, uterine distension, premature placental senescence and oxidative stress, psychosocial stress and racism, and genomic variants at several loci [20,21,22,23,24,25,26,27,28,29,30,31,32,33,34,35,36,37,38,39,40,41,42]. Across these pathologic processes and conditions, inflammation is often a central mechanism triggering preterm labor or prematurely accelerating maturation of parturition pathways [3,43,44]. Although inflammation plays a critical role in the normal delivery process, dysregulated and pathologic inflammation can induce premature decidual and fetal membrane activation. Pathologic inflammation and the various other aforementioned processes thus comprise the ‘upstream’ initiators that confer fetal membrane activation, followed subsequently by the upregulation of ‘downstream’ effectors (e.g., cytokines, chemokines, contraction-associated proteins, prostaglandins, neuropeptides, matrix metalloproteinases, and receptors for inflammatory mediators) [20,45,46,47,48,49,50,51,52]. These pro-inflammatory initiators activate and potentiate a ‘parturition cascade’, leading to cervical shortening and dilation, chorioamniotic membrane rupture, and a contractile state in the myometrium. Most tocolytic agents act to directly suppress uterine contractility (e.g., betamimetics, magnesium sulfate) or inhibit the downstream mediators that stimulate the parturition pathway (e.g., indomethacin, atosiban). New therapeutics in pre-clinical development are shifting focus to address the upstream initiators (e.g., IL-1β receptor antagonists, BSCIs) to confer a broader delay of PTB. When possible, we have addressed specific mechanisms of action for each therapeutic in the parturition cascade.

## 3. The Earliest Tocolytic Agents: Betamimetics

Betamimetics were among the first drugs studied—in the 1960s—for potential use in tocolysis [53]. Some betamimetics, like terbutaline, were tested and approved as anti-asthmatic agents before having any widespread application as tocolytics [54,55]. Off-label use to prevent PTB began with terbutaline being labeled as category B under the prior Federal Drug Administration (FDA) labeling system, which indicated its potential to cause birth defects; category B drugs, however, included prenatal vitamins and use was thought to be routine and safe in pregnancy [54,55]. Terbutaline came under FDA scrutiny in 1997 and was labeled in 2011 as a category C drug; this class indicates that a drug is associated with adverse outcomes in animal studies. Well-controlled human studies, however, are lacking, and the therapeutic benefit may outweigh any risks [54]. Category C labeling of terbutaline was controversial, with many arguing that terbutaline’s efficacy in delaying PTB outweighed its side effects and potential harms [54]. At present, terbutaline still lacks FDA approval for tocolysis but continues to be used in PTB management. In 1977, ritodrine was developed and granted FDA approval for intravenous administration to treat PTB [54,56]. Tocolytic efficacy of oral ritodrine was demonstrated in 1980. In 1995, both oral and IV ritodrine were discontinued and removed from the U.S. market due to adverse cardiovascular effects [56,57]. At present, ritodrine is still used internationally as a tocolytic. Hexoprenaline and salbutamol are other betamimetics that were never marketed in the U.S. but that have limited use internationally [58,59].

Betamimetics function as beta-2 adrenergic receptor agonists, binding to receptors on cell membranes of smooth muscle tissues, which ultimately inhibits myometrial contractions. Activation of these receptors stimulates G_s_ proteins that bind to adenyl cyclase, subsequently increasing cAMP concentrations [60]. Increased cAMP leads to activation of cAMP-dependent protein kinase (PRKA) which phosphorylates and inhibits myosin light chain kinase (MLCK) [60]. Inactivated MLCK is unable to phosphorylate smooth muscle myosin, which prevents interaction with actin, reducing contractile force in smooth muscle tissues [60]. PRKA can also inhibit phospholamban, an inhibitor of SERCA channels [60]. This allows for increased calcium uptake and reduced intracellular calcium, leading to less MLCK activation, calcium-calmodulin interaction, and cross-bridge interactions between myosin and actin [60]. Betamimetic tocolytics reduce uterine contractions through binding to beta-2 receptors located on myometrial cell membranes, slowing the progression of labor.

Despite their use as tocolytics, betamimetics have not been shown to reduce perinatal death or complications of neonatal prematurity compared with placebo [60]. Some studies comparing tocolytic efficacy and adverse outcomes between ritodrine and terbutaline have found no significant differences between them, though other trials have demonstrated an increased efficacy of oral terbutaline compared with oral ritodrine [60,61,62,63,64]. Despite generally similar side effect profiles, numerous studies have described an increased risk of hyperglycemia with oral terbutaline administration and tachycardia following intravenous ritodrine when compared with placebo [59,60,65]. In countries where both agents are available, terbutaline is infrequently used for tocolysis, largely due to a general preference for the safety profile of ritodrine [55,64]. In some instances, however, terbutaline is administered as a single dose subcutaneous injection to examine preliminary uterine reaction to tocolytics or as maintenance therapy following initial treatment with other tocolytics [55,59]. Maintenance therapy with either betamimetic, however, has not been found to further delay PTB or decrease perinatal mortality and morbidity [10,11]. A meta-analysis compared trials of intravenous ritodrine alone with combination therapies of IV ritodrine and other tocolytic agents (e.g., intravenous or oral magnesium sulfate, magnesium gluconate, and indomethacin) [7]. In nearly all trials, combination therapies with ritodrine showed no significant differences in efficacy or adverse effects compared to ritodrine alone [7]. In contrast, other studies have found that combination therapy with progesterone (P4) may allow for increased expression of beta-2 receptors on the myometrium and therefore may influence betamimetic impact [66,67].

Side effects of tocolytic betamimetics are extensive, as beta-adrenergic activity is responsible for a wide range of homeostatic functions. Stimulation of beta-adrenergic receptors upregulates sympathetic pathways, leading to tachycardia, palpitations, chest pain, arrhythmias, tremors, nausea, vomiting, headaches, nervousness, anxiety, and shortness of breath as well as hyperglycemia, hypokalemia, bronchodilation, hypotension, and pulmonary edema [59,60]. Mild beta-1 receptor binding activity contributes to an increased risk of maternal tachycardia [59]. Both terbutaline and ritodrine interact at the maternal-fetal interface and can cross the placenta to induce fetal tachycardia, hypoglycemia, and hyperinsulinism [60]. Recent studies have also suggested that exposure to ritodrine or terbutaline during pregnancy may be associated with an increased risk for autism in the child [57,68]. After adjusting for confounding variables, one study demonstrated that ritodrine administration for delaying PTB has been associated with a moderately increased risk of autism (HR 1.23, 95% CI 1.05–1.47) [57]. It is unclear whether this outcome is due to betamimetic exposure during preterm labor or more generally to PTB and significant fetal exposure to intrauterine inflammation that adversely impacts fetal neurodevelopment [57]. However, in rat models, administration of betamimetics has been associated with adverse neurobehavioral teratogenicity, and terbutaline, specifically, has been shown to alter fetal neurochemistry [57,68].

In a cost-effectiveness analysis comparing tocolytic drug classes, terbutaline and ritodrine were shown to be relatively expensive, costing hundreds of U.S. dollars compared to nifedipine or indomethacin, which are often available for less than 20 U.S. dollars [69]. Betamimetics have been shown to be less effective than numerous other therapeutic agents, and their side effect profile is sufficiently significant to warrant use of other therapeutics when available [60]. Despite the waning use of betamimetics as tocolytic agents in higher-income countries, they continue to be used significantly in countries where other tocolytic agents are not easily available [60].

## 4. Oxytocin Antagonism: Hope for Atosiban

Atosiban, an oxytocin receptor antagonist, was first described in the literature in 1985 and in 1994 was developed specifically as a tocolytic agent [70]. It is currently available for clinical use in Europe but not the U.S. [59,71,72]. Despite a promising maternal safety profile, the FDA failed to approve atosiban for clinical use due to the possible association between atosiban and death in premature infants [73,74]. The FDA also declined to approve atosiban because the studies available at the time were not well designed, ultimately failing to demonstrate improved neonatal outcomes compared to a placebo or other tocolytics [54]. While this decision was largely in accordance with the Kefauver-Harris Amendments, passed in 1962 in response to Germany’s thalidomide scandal, this precedent crippled PTB therapeutic development [75]. The regulatory hurdles resulting from the final edict on atosiban continue to impact the field of PTB therapeutic development and will be expanded upon in a later section of this review (Box 1).

Box 1U.S. regulatory hurdles for PTB therapeutics.
Institutional Review Boards, participants, and physicians have heightened concern when dealing with any drug that has a potential effect on the fetus and on the motherCan people give adequate informed consent during active labor?No templates of successful study design in tocolytic studies, since the only drugs previously approved by the FDA were approved under standards that are now deemed inadequateNo tocolytic drug presented to the FDA has consistently shown neonatal benefit in controlled trialsLack of standardization for disease risk, severity, and progression for common diagnosesSmall market: sales for ‘blockbuster drugs’ in the U.S. for 2004 (~$5 billion) versus annual sales for a tocolytic agent, estimated
at <$500 million


Oxytocin serves numerous roles in the body—namely the promotion of milk letdown for breastfeeding and the stimulation of myometrial contractions during labor. When activated, myometrial oxytocin receptors, which are G-protein-coupled receptors linked to phospholipase C (PLC), activate PLC to trigger a signaling cascade, beginning with increased IP_3_ and diacyl glycerol (DAG), which ultimately leads to calcium release into the cytosol and muscle contraction. Estrogen upregulates myometrial oxytocin receptors to mature the contractile ability of the uterus [76,77]. Although typically produced in the hypothalamus and released from the posterior pituitary, oxytocin is also produced by the uterine decidua during late gestation in preparation for parturition [77]. As an oxytocin receptor antagonist, atosiban blocks this pathway to produce tocolytic effects. The full mechanism, however, has not been determined.

Additionally, atosiban acts as a vasopressin receptor antagonist and may impact prostaglandin F (PGF_2α_) signaling pathways, which normally contribute to the onset and maintenance of labor [78]. These other mechanisms may be involved in atosiban’s tocolytic effects, though further research into these alternative pathways is needed [16]. Atosiban lacks oral bioavailability as a peptide antagonist and is administered subcutaneously or intravenously. Other oxytocin receptor antagonists, barusiban and retosiban, have been studied, but have not been used for tocolysis outside of clinical trials [16].

Atosiban has been shown to significantly reduce uterine contractions during PTB and delay labor by up to 48 h, without conferring significant maternal, fetal, or neonatal adverse effects [8,78,79]. One meta-analysis and multiple other studies have found no significant differences in rates of delivery at 48 h when comparing atosiban with a placebo, betamimetics, or calcium channel blockers (CCBs) [16,80,81]. Data are not available to suggest atosiban improves neonatal outcomes, but its lower discontinuation rate and safer side effect profile have led some studies to conclude that atosiban is preferable to betamimetics [16,80,81]. Like other tocolytic agents, long-term maintenance therapy with atosiban has generally not been shown to delay PTB beyond 48 h, and a meta-analysis suggested maintenance therapy was associated with a significantly higher incidence of injection site reaction when compared with a placebo [9,81]. One study found that, following successful initial intravenous administration of atosiban, continued subcutaneous therapy with atosiban reduced the need for subsequent intravenous atosiban therapy [82]. Combination therapies between intravenous atosiban and other tocolytic agents have demonstrated no significant superior effects compared to atosiban alone [7].

Atosiban is associated with fewer maternal adverse effects than other tocolytic agents, like betamimetics or calcium channel blockers (CCBs) [8]. Maternal side effects are infrequent and include adverse reactions at the site of injection, nausea, vomiting, headache, chest pain, and hypotension [8,16,79]. Atosiban crosses the placenta, but has not been shown to accumulate in the fetus [8]. Compared with a placebo and other tocolytics, Atosiban causes a similar range of possible adverse neonatal effects, and it has not been shown to reduce the incidence of respiratory distress syndrome in premature infants when compared to a placebo [8]. Though betamimetic agents predominate in the global market for tocolytics, atosiban is often considered a drug of choice for tocolysis in high-resource settings where it is available.

## 5. Calcium Channel Blockers (CCBs): Low Cost and Favorable Safety Profile

Nifedipine was first synthesized in 1966 as a coronary vasodilator [83]. Nifedipine and nicardipine (another calcium channel blocker) have been studied as first line tocolytic agents for nearly 20 years but have not been granted FDA approval for use as tocolytics [84].

CCBs act on voltage-gated L-type calcium channels (VGCCs) by binding to the dihydropyridine (DHP) site to block the calcium influx that follows membrane depolarization. Decreased intracellular calcium leads to reduced calcium-calmodulin interactions and reduces activation of MLCK. Similar to betamimetics, decreased MLCK activity inhibits myosin-actin crossbridge formation and reduces muscle contraction. VGCCs are upregulated in uterine myometrial tissue during pregnancy and labor, providing the impetus to study the use of nifedipine and nicardipine in reducing uterine contractions to delay labor [67].

CCBs are used off-label as tocolytic agents and are very popular in the U.S. Certain studies have demonstrated a comparable or increased benefit from CCBs over betamimetics, not only with regards to delay of preterm labor and delivery but also in improving neonatal morbidity and reducing maternal adverse complications [16,84]. Administration of CCBs has not been shown to improve perinatal mortality [16]. Nifedipine is equally efficacious when compared to atosiban, though the lower cost of nifedipine and the impact on reducing neonatal morbidity often makes it the preferred drug of choice [67]. Other CCBs, such as nicardipine, can be administered intravenously or orally; this flexible route of administration is a benefit over nifedipine, which is only orally or sublingually administered [85]. However, nicardipine may have reduced efficacy in prolonging the duration of pregnancy and may produce more side effects in neonates [67,85]. Though nifedipine is not FDA-approved for tocolysis, numerous meta-analyses have shown that CCBs may sometimes effectively delay PTB by up to 7 days, a significant increase in the extent of PTB delay when compared with the 24 h or 48 h delay conferred by most other therapeutics [59,86]. Maintenance tocolysis with nifedipine is ineffective in further delaying labor or reducing neonatal complications when compared to a placebo or observation [12,84].

CCBs and betamimetics have long been used in combination for treatment of cardiovascular diseases, and synergistic effects have been observed in myometrial tissue following combination therapy; interestingly, administration of nifedipine followed by terbutaline confers a synergistic efficacy, but the effect is diminished if administered in reverse order [67]. Combination therapy with these agents is sometimes contraindicated, especially in twin pregnancies and pre-existing maternal cardiovascular disorders, due to increased risk of myocardial infarction and pulmonary edema [67]. Animal studies have demonstrated that pre-treatment with progesterone (P4) can decrease the efficacy of nifedipine; the relationship between these tocolytics is still being assessed [67]. Emerging research on combination therapy with atosiban and nifedipine suggests a strong synergistic relationship, while combination therapy with nifedipine and celecoxib shows a decreased efficacy compared to the use of nifedipine alone [67].

Adverse effects following nifedipine and nicardipine use are typically mild [59]. The most common side effects are headaches, associated with the transient hypotension induced by these drugs. Reflexive tachycardia, anxiety, nausea, vomiting, skin flushing, and palpitations have also been described [67]. Side effects specific to CCB use in tocolysis include severe maternal dyspnea and pulmonary edema, though these manifestations are rare [67]. Nifedipine has not demonstrated any direct adverse fetal side effects [67].

In summary, the extended multi-day delay in labor and the favorable safety profile of nifedipine makes it an appealing agent for tocolysis; the low cost and oral route of administration also allows for implementation of nifedipine in low- and middle-income countries. Like most other PTB therapeutics, CCBs have shown negligible impact when used for maintenance tocolysis. Additional studies on long-term use of CCBs and combination therapy with CCBs and other tocolytic agents is needed [67].

## 6. Nonsteroidal Anti-Inflammatory Drugs as Tocolytic Agents: A Short-Term Prostaglandin Blockade

Non-steroidal anti-inflammatory drugs (NSAIDs), including indomethacin, ibuprofen, and aspirin, have a long history of use in the prevention of PTB; these vasoactive agents were first used as tocolytics in the 1970s [87]. The main mechanism of action of NSAIDs is inhibition of the cyclooxygenase (COX) enzyme, which facilitates the conversion of arachidonic acid to prostaglandins. Prostaglandins are involved in the parturition pathway and facilitate uterine contractions and cervical ripening. Certain NSAIDS, including indomethacin, are thought to inhibit neutrophil activation by suppressing inflammatory stimuli [88]. Indomethacin and ibuprofen reversibly bind to COX, while aspirin binds irreversibly; although each agent binds to both COX-1 and COX-2 isoforms, these drugs have been shown to bind to COX-1 with varying increased selectively [89,90,91]. In pregnancy, both COX-1 and COX-2 are expressed by the myometrium, decidua, and chorioamniotic membranes, and research is ongoing to evaluate the differential expression of those isoforms in these tissues [92,93,94,95,96,97,98,99].

There is little evidence to suggest COX-inhibitors should be prioritized as effective tocolytics when compared with other available therapeutics; however, some studies have indicated a potential use for NSAIDs in preventing PTB [100]. In 2015, a Cochrane review and meta-analysis evaluated the use of indomethacin for the prevention of PTB [101]. In one small study (*n* = 36), indomethacin reduced the risk of PTB before 37 weeks’ gestation (relative risk (RR) 0.21, 95% Confidence Interval (CI) 0.07–0.62) [101]. Indomethacin administration was also associated with a greater gestational age at birth (two studies, *n* = 66; average mean difference (MD) 3.59 weeks, 95% CI 0.65–6.52) and higher birth weight (two studies, *n* = 67 infants; MD 716.3 g, 95% CI 425.5–1007.2) [101]. In this meta-analysis, two trials (total *n* = 70) compared indomethacin to placebo and determined that, with indomethacin, the risk of PTB within 48 h of treatment initiation was reduced, albeit with a wide confidence interval (relative risk (RR) 0.2, 95% CI 0.03–1.28). In trials comparing indomethacin to beta-mimetics, indomethacin reduced the risk of PTB less within 48 h (two trials, total *n* = 100; RR 0.27; 95% CI 0.08–0.96) and PTB before 37 weeks of gestation (two studies, *n* = 80; RR 0.53; 95% CI 0.28–0.99) compared to betamimetics [101]. There was no difference between indomethacin and magnesium sulfate (seven studies, *n* = 792) or CCB (two studies, *n* = 230) in terms of pregnancy prolongation or fetal/neonatal outcomes [101]. A selective COX-2 inhibitor, Rofecoxib, was studied for potential use in tocolysis and ultimately shown to not reduce the incidence of preterm birth [102].

Aspirin is also a COX-inhibitor and has been theorized to be useful in the prevention of PTB by decreasing the incidence of preeclampsia [103]. Recently, meta-analyses and secondary analyses of trials of women taking aspirin for the prevention of preeclampsia have shown a decreased rate of PTB, as preeclampsia is a risk factor for PTB [104,105,106]. A secondary analysis of an RCT examining aspirin for the prevention of preeclampsia showed that in nulliparous, low-risk women, there was a decreased rate of spontaneous PTB < 34 weeks among women who received aspirin (*n* = 2543; AOR 0.46; 95% CI 0.23–0.89) [106]. In a larger RCT comparing aspirin to placebo in nulliparous women, aspirin was associated with a decreased rate of preterm delivery (*n* = 11,976; RR 0.89; 95% CI 0.81–0.98) [104]. In this same study, among those who did deliver preterm, aspirin was associated with a decreased rate of early preterm delivery < 34 weeks (RR 0.75, 95% CI 0.61–0.93) [104]. Low-dose aspirin prophylaxis has been indicated for reducing risk of preeclampsia in women with particular high-risk factors; however, further evaluation of aspirin for use as a tocolytic is necessary to affirm current findings.

Maternal side effects of NSAID use generally depend on length of time of treatment. Serious side effects are uncommon if NSAIDS are used for less than 48 h. Common maternal reactions include nausea and heartburn but can also involve gastrointestinal bleeding and prolonged bleeding time [107,108]. Prolonged treatment with NSAIDS can also lead to renal injury [109]. In the 2015 Cochrane review, COX-inhibitors were found to have fewer maternal side effects compared to betamimetics and decreased cases of treatment cessation due to maternal adverse effects. Compared to magnesium sulfate, COX-inhibitors also had fewer maternal adverse effects [101]. Maternal contradictions to indomethacin include chronic renal or hepatic disease, peptic ulcer disease, and coagulation disorders [110].

NSAIDs cross the placenta and can result in premature closure of the ductus arteriosus leading to neonatal pulmonary hypertension. Constriction of the ductus arteriosus due to maternal NSAID use occurs via inhibition of prostacyclin and prostaglandin E2, which maintain vasodilation of the duct [111]. Additionally, NSAIDs may diminish prostaglandin inhibition of antidiuretic hormone and affect fetal renal blood flow, typically resulting in dose-dependent and reversible oligohydramnios [112]. These adverse effects are generally associated with use of COX-inhibitors for longer than 48 h before 32 weeks’ gestation and for any length of time after 32 weeks [111,113,114,115]. The 2015 Cochrane review noted that data were insufficient to determine fetal safety [101]. As many NSAID-related side effects appear to be dose-dependent, low-dose aspirin and other NSAIDs are considered safe throughout much of pregnancy. Notable fetal complications, however, arise from the inappropriate use of higher-dose NSAIDs in the third trimester.

Despite a near fifty-year history of use in the management of PTB, there is a relative paucity of studies on NSAID use in tocolysis, particularly with regards to combination therapies or as a maintenance tocolytic. The significant adverse and sometimes life-threatening fetal complications induced by these drugs warrant careful administration of NSAID therapeutics later in pregnancy and present an important challenge to navigate when studying these agents in future PTB clinical trials.

## 7. Magnesium Sulfate: A Classic Tocolytic, Called into Question

As early as the 1940s, magnesium sulfate was shown to promote uterine quiescence [116,117]. It was originally studied for potential use in reducing abnormal uterine contractions underlying severe dysmenorrhea [116]. In 1977, magnesium sulfate was first described for use in tocolysis [118]. In 1983, a case series of 355 patients demonstrated a dose-dependent association between magnesium sulfate and tocolysis [119]. It has since become one of the most commonly used therapeutics for delaying PTB [120,121].

The mechanism for magnesium sulfate as a tocolytic has not been fully determined. Physiologically, magnesium regulates calcium uptake, which impacts muscle contraction and neuronal activity. This calcium regulation is thought to be due to both an extracellular effect of decreasing the intracellular calcium stores as well as an intracellular effect of blocking the release of intracellular calcium via IP_3_ receptor/channels [120,122,123,124,125]. Magnesium sulfate also appears to impact the binding and distribution of calcium in smooth muscle, including uterine tissues, reducing the frequency of cellular depolarization to potentially reduce myometrial contractions [126]. Interestingly, magnesium sulfate must be kept at suprapharmacologic levels (4–10 mmol, 8–16 mEq/L) in vitro in order to effectively inhibit cyclic uterine activity [123].

Whether magnesium sulfate delays PTB significantly in comparison to other available tocolytics is unknown [127,128]. Comparisons of different studies of magnesium sulfate are challenged by the complexity of underlying PTB pathophysiology, sub-therapeutic dosing in some studies, and generally small sample populations [119]. In a meta-analysis of 19 randomized clinical trials, magnesium sulfate was not shown to reduce the frequency of delivery within 48 h when compared with a placebo (RR 0.75, 95% CI 0.54–1.03), and there was no significant reduction in delivery before 37 weeks’ gestation (RR 1.18, 95% CI 0.93–1.51) [120]. Magnesium sulfate has also not been shown to provide improvements in neonatal morbidity or mortality [120]. Another meta-analysis of 95 trials of varying tocolytics (31 studies, encompassing 2653 patients, focused specifically on magnesium sulfate) found that magnesium sulfate was comparable to prostaglandin inhibitors in its probability of effectively delaying PTB by up to 48 h [129,130]. This meta-analysis, however, noted that other tocolytics (e.g., prostaglandin inhibitors and calcium channel blockers) provide a more effective delay of PTB, with better neonatal outcomes and safer side effect profiles. The 1983 case series study demonstrated significant differences in effective tocolysis among women who received varying doses of magnesium sulfate, with high doses associated with the greatest tocolysis [119,131]. More recent studies comparing high- and low-dose intravenous magnesium sulfate found no significant differences in fetal or neonatal death and were unable to verify a dose-dependent impact on delaying PTB when compared with a placebo [129]. Compared to short-term administration, long-term administration of magnesium sulfate is associated with the development of fetal osteopenia [119]. Despite conflicting and generally low-power studies of the efficacy of magnesium sulfate as a tocolytic, there is a known benefit to administering magnesium sulfate for fetal neurological protection to pregnant people at risk for PTB [128,129,132,133,134,135]. Magnesium sulfate supplementation may also provide protection against preeclampsia—a known risk factor for PTB—though more research is needed for confirmation [136].

The adverse effects of magnesium sulfate in mothers include respiratory arrest and pulmonary edema, nausea, flushing, gastrointestinal disturbance, lethargy, and blurred vision [137]. Common adverse effects in neonates include lethargy, poor suckling, and delayed time to onset of established respiration [138]. Prolonged magnesium sulfate administration has been suggested to increase risk of fetal hypocalcemia and osteopenia, as magnesium sulfate is known to cross the placenta and cause fetal hypermagnesemia; fetal bone calcification can be inhibited by magnesium sulfate, which may lead to changes in bone thickness [119,139]. High doses of magnesium sulfate have also been suggested to contribute to increased risk of fetal and neonatal death [140].

Magnesium sulfate has long served as one of the more well-known drugs employed for PTB management. There is, however, a significant lack of data to support therapeutic efficacy in delaying PTB, despite magnesium sulfate’s extensive clinical use as a tocolytic agent. Both efficacy and adverse effects of magnesium sulfate appear to be at least partially dose-dependent. As other therapeutic classes gain prominence in the field of PTB management, magnesium sulfate may eventually decline in use.

## 8. Addressing Infection: Trials of Antibiotics to Prevent PTB

Infection and inflammation are frequent causes of early PTB, and infections are typically polymicrobial [24,141,142,143,144]. Numerous studies have sought to determine the specific associations between preterm labor and infectious states in order to elucidate the potential utility of antibiotic administration in preventing or delaying PTB. Multiple randomized clinical trials have failed to demonstrate a benefit from antibiotic therapy for women in preterm labor with intact membranes [145]. A key study in this area was the ORACLE II randomized controlled trial, which addressed the question of whether administration of broad-spectrum antibiotics might prolong pregnancy and prevent neonatal morbidity in women with preterm labor and intact membranes, who had no evidence of an amniotic fluid infection [146]. Antibiotic administration in the absence of an evident infection has long been understood to carry numerous risks, which include detrimental impacts on an individual’s microbiome, increasing risk for antibiotic-resistant neonatal sepsis, and antibiotic resistance in the mother if undertreated or inappropriately treated [147]. In the ORACLE II trial, women were randomized to either co-amoxiclav, erythromycin, co-amoxiclav and erythromycin, or a placebo for 10 days or until delivery. Antibiotic therapy did not prolong pregnancy or improve neonatal outcomes; higher rates of necrotizing enterocolitis (0.6% vs. 0.3%) were observed in the neonates exposed to co-amoxiclav, but the data was not statistically significant [145,146]. In addition, higher rates of neonatal mortality and cerebral palsy were observed in infants and children who received both macrolide and beta-lactam antibiotics [145,146]. When the children initially included in the ORACLE II trial were reevaluated at 7 years of age, there were higher rates of cerebral palsy in children of women who received both antibiotics (3.3% vs. 1.7% for erythromycin, 3.2% vs. 1.9% for amoxiclav) and higher rates of functional impairment in those who received erythromycin (42.3% vs. 38.3%) [148]. Thus, the use of antibiotics as a PTB therapeutic is not recommended in GBS-negative women due to the lack of benefit in preventing PTB and concern regarding an elevated risk for the development of neonatal necrotizing enterocolitis or other adverse effects conferred by unnecessary antibiotic administration.

Preterm premature rupture of membranes (PPROM) is a major cause of PTB, and women with PPROM are highly likely to have a subclinical or overt intra-amniotic infection [149,150,151,152]. In contrast to the treatment of PTB with intact membranes, antibiotic therapy as a means of tocolysis is recommended and routinely used for women with PPROM in high-income countries [153]. There are, however, notable deficiencies in clinical knowledge regarding which antibiotic agents should be administered as well as the respective regimens and dosages to adequately address PTB in the setting of PPROM. Nevertheless, intravenous ampicillin and erythromycin, followed by oral amoxicillin and erythromycin, has been shown to significantly prolong pregnancy in GBS negative women and lower rates of neonatal respiratory distress (40.5% vs. 48.7%) and necrotizing enterocolitis (2.3% vs. 5.8%) in those with and without GBS colonization [154]. The use of broader spectrum antibiotics like co-amoxiclav is not recommended, however, due to an increased rate of neonatal necrotizing enterocolitis, similar to the findings in women with PTB and intact membranes [146,155]. In patients colonized with GBS, latency antibiotics were also associated with significantly decreased neonatal sepsis (8.4% vs. 15.6%) and pneumonia (2.9% vs. 7%) [154]. A review of 13 randomized controlled clinical trials that included more than 6000 women with PPROM before 37 weeks’ gestational age found that the use of latency antibiotics was associated with decreased rates of chorioamnionitis (RR 0.66, 95% CI 0.46–0.96), neonatal infection (RR 0.67, CI 95% 0.52–0.850), oxygen requirements (RR 0.88, 95% CI 0.81–0.96), and abnormal cerebral ultrasound findings (RR 0.81, 95% CI 0.68–0.98) [156]. Based on this evidence, narrower spectrum antibiotics are routinely given to women with PPROM and have shown neonatal benefit. Conflicting data, however, complicate recommendations for employing a broader scope of tocolytic agents in managing patients with PPROM [157,158]. Numerous studies have demonstrated that tocolytic (e.g., betamimetics, CCBs, and COX-inhibitors) administration does not reduce perinatal mortality in women with PPROM and is instead associated with a potentially increased risk of chorioamnionitis as well as increased risk for subsequent need of respiratory support for neonates when compared with a placebo [157,158]. Therefore, although there is limited data to indicate the utility of antibiotics in managing PPROM-induced PTB, the clinical benefit of administering a broader range of therapeutics to delay PTB in patients with PPROM remains unproven.

Bacterial vaginosis (BV) is a complex, heterogenous vaginal infection associated with PTB [24,142,159,160,161,162,163,164,165]. Unfortunately, antibiotic treatment trials of BV have generally not reduced PTB [166,167,168,169]. The US Preventative Task Force Services released updated recommendations in 2020 regarding screening and treatment of BV in pregnancy. After reviewing results from 13 randomized clinical trials, they recommend against treatment of asymptomatic BV in pregnancy, as treatment does not seem to decrease adverse obstetrical outcomes including PTB, low birth weight, or PPROM [170,171,172]. The benefits of treating high-risk individuals with a history of PTB is less clear [170]. There are numerous problems associated with prior trials of BV in pregnancy to prevent PTB, which may have confounded these studies. First, BV treatment in pregnancy may be too late to prevent microbial trafficking into the uterus; pre-conception treatment may be necessary to optimize outcomes. The antibiotics tested may be ineffective against the many uncultivable pathogens associated with BV [173,174]. Finally, BV treatment may shift the vaginal microbiota from one BV-associated microbiota profile to another, and many clinical trials did not follow changes in the microbiota over time [175]. Understanding the gene-environment interaction of BV and PTB could broaden the approach to treatment or prenatal care. Pregnant people with BV who had certain genotypes of PRKCA, FLT1, and IL-6 genes were more likely to deliver preterm than people without BV [176]. Although screening and treatment of BV is not currently recommended to prevent PTB, research in this area continues to evolve.

Screening and treatment of sexually transmitted infections (STIs; e.g., *Chlamydia trachomatis*, *Neisseria gonorrhoeae*, *Trichomonas vaginalis*) is recommended to reduce the neonatal risks associated with maternal infection and reduce STI transmission in the community. Antibiotic treatment of STIs does not appear to prolong pregnancy and prevent PTB [177]. Likewise, treatment of asymptomatic urinary tract infections with antibiotics does not lower rates of PTB [178,179,180]. Nevertheless, urinary tract infection treatment is recommended to decrease rates of other high-risk outcomes, such as pyelonephritis and low birth weight. Metronidazole should be offered to people with a symptomatic *T. vaginalis* infection in pregnancy to relieve maternal symptoms and prevent spread of STIs. Although metronidazole is safe for use in the first trimester of pregnancy, there is concern that treatment of *T. vaginalis* with metronidazole in the late second or third trimester of pregnancy may increase rates of PTB [181,182]. Vaginal candidiasis has not been shown to be a risk factor for PTB, and treatment appears to be safe in pregnancy [183]. Overall, the treatment of STIs and UTIs has not been shown to reduce the rate of PTB, but treatment of UTIs in early pregnancy confers both maternal and neonatal benefit.

## 9. Progesterone (P4) Analogs

The first studies on the use of P4 to prevent PTB were in the 1960s, demonstrating a potential protective effect of P4 administration [184,185]. Studies of P4 as a PTB therapeutic dwindled throughout the 1990s, until two studies in 2003 reinvigorated interest in P4 as a viable preventative strategy [184,186,187,188]. In 2011, the FDA approved the use of hydroxyprogesterone caproate (17-OHPC) to reduce the risk of PTB [184]. The approval of 17-OHPC for use as a preventative agent marked the first drug approved for use in pregnancy in over 15 years [184]. However, significant controversy surrounds the use of P4 to prevent recurrent spontaneous PTB, and the FDA has proposed the withdrawal of this approval after post-market studies failed to verify clinical benefit [189]. Additional studies of P4 for use in tocolysis are ongoing (Table 2).

P4 is a pro-gestational hormone, which maintains uterine quiescence for most of pregnancy by inhibiting the transcription of labor-associated genes in the myometrium. These genes include *gja1* (Connexin 43), *otr* (oxytocin receptor), and *ptgs2* (cyclooxygenase), which encode uterine activation proteins; *nfkb* (nuclear factor kappa-B), *ilb1b* (IL-1β, *ccl2* (MCP-1), and *cxcl8* (IL-8), which encode proteins involved in cytokine and chemokine signaling. Myometrial activation and the initiation of labor requires withdrawal of P4, which in most viviparous species is achieved by a fall in peripheral P4, a key trigger of parturition. In contrast, circulating P4 levels remain elevated in women at term [190]. Yet, blocking P4 action at any time during human pregnancy by administration of the specific antagonist, RU486, induces preterm labor. Therefore, it has been hypothesized that a ‘functional withdrawal’ of P4 occurs in the human myometrium, whereby uterine smooth muscle cells become resistant to the pro-gestational actions of P4, inducing labor onset [191,192].

Although the molecular mechanism by which P4 controls the timing of labor onset is not well understood, recent studies have shed light on how a functional P4 withdrawal might occur in pregnant people. Recently, it was reported that during gestation, P4 signals predominantly through its nuclear receptor PRB, suppressing the transcription of labor-associated and inflammatory genes and promoting myometrial relaxation [193]. Further study found the truncated PR isoform, PRA (when not bound to P4, ‘un-liganded’), antagonizes the action of PRB. At the end of gestation, the increased nuclear abundance of PRA relative to PRB may be an important factor in antagonizing the signaling of P4, which would promote CAP gene expression and labor initiation [193,194]. Importantly, despite the abundance of P4 in maternal blood during term labor, intracellular P4 concentration in myometrial cells decreases due to an increased expression of the P4-metabolizing enzyme 20α-hydroxysteroid dehydrogenase (AKR1C1/20α-HSD), which converts P4 into its inactive metabolite, 20alpha-hydroxyprogesterone (20α-OHP). Thus, it is possible that catabolism of P4 in myometrial cells leads to a local withdrawal of P4, the un-liganding of PRA from P4, which in turn switches PRA to an activator of CAP gene transcription, triggering labor onset. In summary, the P4/PR signaling pathway in the myometrium may represent a target for the development of novel therapeutic agents for PTB prevention. Administration of synthetic progestins that are not metabolized by 20α-HSD may maintain liganding of P4 to PRB and suppress the transcription of genes that otherwise would induce labor (Figure 1).

While P4 is required for the maintenance of human pregnancy, research on its use as a prophylactic therapeutic to prevent PTB has produced contradictory data [186,187,195,196,197,198,199]. In 2003, a small double-blind trial found that 17α-hydroxyprogesterone caproate (17-OHPC, weak progestin) significantly reduced the risk of PTB in people with a prior PTB [187]. However, other studies failed to find evidence that P4 can prevent recurrent PTB. For instance, larger trials have recently shown that the effectiveness of P4 is limited [200,201]. A double-blind, randomized, placebo-controlled trial of vaginal P4 prophylaxis in high-risk women with a previous spontaneous PTB was not associated with reduced risk of PTB or composite neonatal adverse outcomes [196]. A 2017 meta-analysis showed that administration of vaginal P4 to asymptomatic women with a twin gestation and a sonographic short cervix in the mid-trimester reduced the risk of PTB as well as neonatal mortality and morbidity [202]. However, only one study conducted in women with singleton pregnancies and a mid-gestation sonographic cervical length greater than 25 mm resulted in a positive outcome for prophylactic therapy against PTB using a vaginal suppository with micronized P4, with others showing no effect [203,204].

A therapeutic strength of P4 is the excellent safety profile in pregnancy, underlying its FDA approval for use as a PTB therapeutic [184,188]. Endogenous P4 naturally induces a number of systemic side effects, including headache, moodiness, and loss of libido. These side effects are significantly reduced with the use of synthetic P4 [188,205]. Micronized P4 has been shown to eliminate these adverse effects entirely [188]. No risk to the neonate has been associated with P4 administration [188,206]. Other P4 formulations may have greater efficacy in preventing PTB than 17-OHPC. R5020 binds PRs with high affinity and is not a substrate for metabolism by 20α-HSD; a pilot study in pregnant rats was associated with inhibition of preterm labor [207]. R5020 or other steroidal or non-steroidal P4 analogs may represent an innovative approach to PTB prevention in people at risk for PTB.

## 10. Inhibiting Inflammation: Use of Broad-Spectrum Chemokine Inhibitors

Numerous studies over the past decade have focused on developing anti-inflammatory agents that target specific chemokines and chemokine receptors underlying pathologic inflammation in certain human diseases. Successes in the development of BSCIs for human immunodeficiency virus (HIV) treatment and prevention of cancer metastasis have spurred recent interest in studying the use of BSCIs in preventing or delaying PTB in high-risk pregnant women [208,209,210]. The exploration of BSCIs for use in addressing PTB is currently limited to animal models [210,211].

Leukocytes are an active component of the maternal immune system; therefore, they can provide relatively accessible means to interrupt the inflammatory pathway that leads to labor initiation. Studies indicate that circulating maternal leukocytes are not activated during most of gestation but become primed and migrate to the uterus before term labor or prematurely due to an infection process [212,213,214,215]. Whether these leukocytes act to help induce labor or represent a consequence of parturition is an active debate [216]. Chemokines mediate the recruitment of immune cells from the peripheral circulation to the site of inflammation by attracting and activating specific leukocyte subsets. There are approximately 50 chemokines that can bind to over 20 distinct chemokine receptors [217]. Blocking one specific receptor may prevent the actions of multiple chemokines, as they induce peripheral leukocyte infiltration into the target tissues. Importantly, the simultaneous blockage of multiple chemokines has not been associated with acute or chronic toxicity, signifying a possible new approach to developing potential PTB therapeutic candidates.

A BSCI can block multiple chemokine signaling pathways [218]. BSCIs specifically bind to the cell-surface somatostatin receptor type 2 (SSTR2) [208,219,220,221,222,223,224,225]. Somatostatin is a cyclic neuropeptide hormone that functions as a mediator between the nervous and the immune systems and regulates the immune response [226,227,228,229,230]. In addition to its role in inflammation, recent studies have explored the role of somatostatin and SSTR2 in myometrial cell contractility [231]. In a porcine model, an experimental *Escherichia coli* infection led to increased uterine expression of SSTR2 and an increased amplitude of somatostatin-stimulated uterine contractions [231]. Importantly, SSTR2 antagonists prevented the increase in contraction amplitude, suggesting a role for SSTR2 in mediating uterine contractility [231]. BSCIs are partial agonists of SSTR2 and inhibit pathways generated by multiple chemokine receptors without affecting classical SSTR2 agonist signaling. Mechanisms of BSCI action are not well understood, but it has been hypothesized that partial agonism of BSCI to SSTR2 enables binding to an inflammatory site, which inhibits directional signals from the receptor; consequently, cells become effectively ‘blind’ to the chemokine gradient. The ability of BSCIs to improve the outcome of many different inflammatory diseases (e.g., allergic asthma, surgical adhesion formation, rheumatoid arthritis, HIV replication, and endometriosis) has been demonstrated in animal models [208,219,220,221,222,223,224]. Targeting chemokine signaling to prevent pathologic uterine activation in high-risk pregnant people may represent a useful therapeutic approach for PTB prevention.

The first in vivo studies demonstrated that prophylactic administration of the BSCI compound ‘BN83470′ was able to prevent infection (LPS)-induced PTB in pregnant mice by blocking multiple inflammatory pathways in the uterus and leukocyte infiltration into uterine myometrium (Figure 2) [211]. As a proof of principle in a nonhuman primate model of Group B Streptococcus (GBS)-induced preterm labor, a novel BSCI compound ‘FX125L’ inhibited preterm labor and suppressed the cytokine response (Figure 2) [210]. Antibiotics were not administered in these experiments, and the GBS infection progressed and invaded the amniotic cavity and the fetus. Despite this invasive GBS infection, BSCI prophylaxis was associated with significantly lower cytokine levels in the amniotic fluid, fetal plasma, lung, and brain, demonstrating its power in suppressing an inflammatory response. However, this result also highlights the danger of administering a chemokine or cytokine inhibitor in the setting of an intrauterine infection without concomitant antibiotic administration; preterm labor may be inhibited, while an intrauterine infection silently invades and injures the fetus. Nevertheless, these data represent an important proof of principal that BSCIs may be highly effective in preventing PTB in humans. Current studies in animal models have suggested that BSCI compounds do not exhibit any significant fetal toxicity. Further study, particularly in human clinical trials, is required to elucidate the effect of BSCI on the fetal immune response and fetal development [210,211].

## 11. Targeting the Prostaglandin F2 Alpha Receptor: OBE-022

The key role of prostaglandins (particularly PGE_2_ and PGF_2α_) in the initiation and progression of labor has made them a target for therapeutics to prevent PTB [232]. Activation of the PGF_2α_ receptor stimulates myometrial contractions and upregulation of matrix metalloproteinases, which leads to cervical ripening and the rupture of membranes [233,234,235,236]. Since PGE_2_ is the primary fetal prostaglandin, selective inhibition of PGF_2α_ has been proposed to provide effective tocolysis, with reduced fetal morbidity [237].

OBE-022 (ebopiprant) is a highly potent, competitive, reversible inhibitor of the prostaglandin F 2-alpha (PGF_2α_) receptor [238]. OBE-022 has been studied in both human tissues and pregnant animal models and is in Phase II clinical trials. In preclinical studies, OBE-022 inhibited human myometrial contraction in vitro and was more effective in combination with nifedipine or atosiban [238]. In an animal model, OBE-022 showed similar activity to nifedipine for reduction of parturition as well as synergistic effects when used together with nifedipine, without evidence of fetal ductus arteriosus constriction, impairment of fetal renal function, or platelet inhibition [238]. A Phase I trial was performed, which demonstrated that OBE-022 was well tolerated, with no significant side effects in the postmenopausal female population [239,240]. An additional Phase I trial was performed on non-pregnant premenopausal women and demonstrated that OBE-022 was well tolerated alone as well as in combination with nifedipine, magnesium sulfate, atosiban, and betamethasone [241]. Based on the favorable results in the Phase I trials, OBE-022 is currently under investigation in the PROLONG trial, an ongoing Phase 2a randomized, double-blind, placebo-controlled proof-of-concept study in pregnant persons with spontaneous preterm labor at 24–34 weeks’ gestation.

OBE022 is also currently being studied for potential side effects similar to indomethacin, with particular focus on closure of the ductus arteriosus, impaired fetal renal function, and inhibition of platelet aggregation [238]. In preclinical studies, OBE-022 was not shown to increase these fetal adverse effects in rat models [238]. Some of the first studies in humans on the potential adverse effects of OBE-022 found favorable pharmacokinetic characteristics and no significant impact on prolonging the QT interval among a cohort of healthy postmenopausal women [239,240].

Additional study is required to examine the effects of co-administration of OBE-022 with other tocolytics [238]. Preliminary data from this study (and others) suggest an encouraging potential for the approach of selective targeting of prostaglandin receptors. Potent tocolytic effects, with reduced adverse fetal impact, may make OBE-022 a promising new therapeutic for delaying PTB [238].

## 12. Interleukin-1 Receptor Antagonism: Kineret and Rytvela

Early inflammatory events have become an intriguing target for delaying or blocking PTB while protecting the fetus from harmful inflammation [242,243,244,245,246]. A significant upstream target is IL-1β, a pro-inflammatory cytokine thought to be the apex initiator of the parturition cascade (Figure 3) [245]. IL-1β has been shown to induce preterm labor in a mouse model via intrauterine injection as well as in a nonhuman primate model when administered into the amniotic fluid [245,247].

Therapeutic inhibition studies have traditionally involved screening for ligands that bind to the natural ligand’s receptor site (orthosteric binding site). Kineret, canakinumab, and rilonacept are large protein orthosteric antagonists to IL-1β that are approved for clinical treatment of inflammatory disorders but have shown limited efficacy for preventing PTB in animal models [245,248,249]. Of particular interest is kineret, a recombinant version of the endogenous IL-1 receptor antagonist (IL-1RA). In rodent, sheep, and nonhuman primate models, systemic administration of kineret at standard (low) doses reduced IL-1β- and LPS-induced fetal inflammation and injury, but much higher doses were required to inhibit inflammation in the uteroplacental unit and prevent PTB [245,246,248,249,250,251]. Conversely, standard doses of kineret reduced inflammatory signaling at uteroplacental tissues when directly administered to the amniotic fluid in nonhuman primates and primary cultures of human amnion and chorion [252,253]. This discrepancy is likely attributed to the large molecular size of kineret, which may have limited bioavailability to the placenta. As an orthosteric antagonist, kineret may induce undesired side effects by exerting nonspecific inhibition of all IL-1β signaling pathways, including the immunoregulatory NF-κB pathway [254]. Indeed, kineret, canakinumab, and rilonacept have been associated with injection site reactions, upper respiratory tract infections, vertigo, gastrointestinal disorders, and immune suppression [255,256]. Given these limitations, the alternative is to identify ligands that bind to remote allosteric sites with greater selectivity by affecting the conformational dynamics of the receptor and biasing receptor signaling pathways [257,258,259].

Rytvela is an example of an allosteric inhibitor of the IL-1R. Rytvela was designed to target the structural region of the IL-1R adjacent to the extracellular juxtamembranous domain of the IL-1RacP (accessory protein), because of its importance to tyrosine kinase receptor activation [260,261]. Rytvela is a 7-amino-acid peptide (each letter refers to the amino acid nomenclature), which was made resistant to hydrolysis by using all *d*-amino acids. Rytvela has been shown to potently inhibit IL-1β-induced PGE_2_ generation, whereas a scrambled peptide (verytla) was ineffective [262]. Furthermore, Rytvela specifically inhibits IL-1R but not homologous IL-1 family cytokines IL-18 or IL-33, the latter of which shares IL-1RacP; acute effects of other pro-inflammatory cytokines (TNFα, IL-6) are also not affected. In contrast to kineret, rytvela does not inhibit IL-1β-induced NF-κB activation and dependent monocyte phagocytosis, preserving immunosurveillance. However, rytvela does inhibit RhoA kinase, p38 and JNK phosphorylation, which inhibits IL-1β from generating more IL-1β—a mechanism that likely leads to blocking PTB but requires further study [243,244,245]. Hence, consistent with IL-1β-induced inflammation independent of NF-κB, rytvela preserves the NF-κB pathway [245,263,264,265]. Rytvela exhibits anti-inflammatory properties in several in vivo disease models (e.g., acute dermatitis, chronic rheumatoid arthritis, acute bowel inflammation, ischemic retinopathy and encephalopathy, and degenerative inflammatory retinopathies) [262,266,267,268].

In rodent models, rytvela has been shown to markedly reduce the uteroplacental surge in inflammatory cytokines and inhibit PTB induced by sterile inflammation from either IL-1β or bacterial-like inflammation triggered by lipoteichoic acid or lipopolysaccharide [245,246]. In these models, rytvela also yielded significant improvements in fetal outcomes. These included reductions in inflammatory signals in the plasma, brain, lungs, retina, and intestines and the associated protection of the fetal brain parenchyma, lung alveolarization, retinal vascular development, and gut mucosal integrity [245,246,266,269,270,271,272]. Rytvela will soon be entering a formal pre-clinical phase for toxicity evaluation.

## 13. Nanoparticles

Nanocarrier systems are aimed at improving the safety profile of tocolytics, and where uterine targeting is included, also improving tocolytic efficacy. These are potentially deliverable benefits, as targeted delivery increases the proportion of a drug that reaches a target tissue while simultaneously reducing localization to non-target tissues. This reduces the amount of drug required to achieve therapeutic efficacy and reduces off-target side effects, which together improve patient safety [273].

Early applications of nanoparticles for improving tocolysis utilized non-targeted liposomes, which are organic, nano-scale lipid vesicles that can be loaded with a broad array of drugs. In 2015, liposomal administration of indomethacin was tested as a method of reducing placental transfer and avoiding adverse effects on the fetus [274,275,276]. Studies in mice showed strong liposome localization to the pregnant uterus, weak localization to the placenta, and no liposomal transfer to fetuses [274]. Moreover, liposomal encapsulation significantly reduced indomethacin fetal transfer (7.6-fold) [274]. The study did not examine liposome localization to other maternal tissues; however, it provided promising insight into the potential benefits of nanoparticle-facilitated tocolysis.

The first targeted nanoparticles for improving tocolysis were reported in 2015, which were nanoliposomes coated with an antibody that recognized an extracellular domain of the oxytocin receptor (OTR; Figure 4) [277]. The rationale was that the high levels of OTR expression on uterine myocytes during pregnancy could serve as a means of targeting therapeutics to the pregnant myometrium [278,279].

Using spontaneously contracting strips of pregnant human myometrium suspended in organ baths, OTR-targeted nanoliposomes were shown to either abolish or significantly enhance contractions when they were loaded with contraction-blocking or contraction-enhancing drugs, respectively [277,280]. This demonstrated the versatility of the approach. Soon after, the same group reported on the in vivo biodistribution of OTR-targeted nanoliposomes in pregnant mice and evaluated their application toward delivering indomethacin for preventing lipopolysaccharide (LPS)-induced PTB [281]. Less than two months later, a complementary analysis by another lab reported on the benefits of administering indomethacin via OTR-targeted nanoliposomes [282]. These two studies provided compelling proof-of-concept data that demonstrated effective targeting of the pregnant mouse uterus in vivo, enhanced drug delivery to the pregnant mouse uterus, reduced fetal transfer of the drug, and significant reductions in rates of LPS-induced PTB. A key difference between the two studies’ OTR-targeted nanoliposomes was their respective OTR targeting moiety, with the former utilizing antibody targeting and the latter utilizing the OTR antagonist peptide, atosiban [281,282]. In a comprehensive in vitro comparison of the two systems, no difference was found in OTR binding, stability, or cellular toxicity [283]. Moreover, both types of OTR-targeted nanoliposomes were shown to be internalized by clathrin- and caveolin-mediated endocytosis, which provided valuable insight into the mechanism of action of OTR-targeted nanoliposomes.

Vaginally administered nanoformulations have also emerged as a promising avenue for preterm birth therapeutics. Uterine efficacy of vaginally administered drugs is facilitated by the uterine first-pass effect; however, vaginally administered drugs must first penetrate the cervicovaginal mucus [284]. To facilitate this, mucus-penetrating particles were developed; they are approximately 200 nm in diameter, with a net neutral surface charge, making them mucoinert (non-adhesive to mucus) and able to penetrate the cervicovaginal mucus [285]. These mucoinert nanosuspensions have been shown to improve the vaginal administration of progesterone and prevent progesterone withdrawal (RU486)-induced preterm birth in mice, while avoiding stimulating myometrial expression of inflammatory cytokines [286]. Other studies have demonstrated successful vaginal administration of the histone deacetylase inhibitors trichostatin-A (TSA) and suberoylanilide hydroxamic acid (SAHA) via mucoinert nanosuspensions [287]. Nanosuspensions of TSA in combination with progesterone had anti-inflammatory effects on myometrial gene expression and were effective in significantly reducing rates of LPS-induced preterm birth in mice, whereas nanosuspensions of progesterone alone were not effective [287].

Similarly, nanoformulations of the sphingosine kinase (SphK) inhibitor (SKI II) were developed, as SphK inhibition was previously shown to prevent LPS-induced preterm birth in mice [288]. To facilitate the use of SKI II, which has extremely low aqueous solubility, it was incorporated into a self-nanoemulsifying drug delivery system (SNEDDS), which is an isotropic mixture of oil, surfactant, and solvent that forms a stable nanoemulsion when dispersed in aqueous media [289,290]. The SNEDDS increased SKI II solubility over 500-fold, and when vaginally administered, it was effective in significantly reducing rates of LPS-induced preterm birth in mice [289].

Nanoparticle-based systems for improving tocolysis have yet to progress beyond preclinical studies. As such, there are only limited data available from which to speculate on a safety profile respective to mothers or their neonates. Nonetheless, the data generated thus far is promising. For liposome-based systems, OTR-targeted nanoliposomes showed no significant impact on myocyte viability in vitro, while ex vivo, drug-free, OTR-targeted nanoliposomes had no effect on contracting pregnant human myometrial tissue strips, and contractions resumed when drug-loaded OTR-targeted nanoliposomes were washed away [281,283]. In vivo, OTR-targeted nanoliposomes produced no adverse effects upon repeated intravenous administration to pregnant mice [281]. The nanoliposomes were not detected in the maternal brain, kidneys, lung, or heart and were only detected at low levels in mammary tissue and placentae [281,282]. Neither of the landmark studies detected fetal transfer of non-targeted or OTR-targeted nanoliposomes [281,282]. For the vaginal nanoformulations, nanosuspensions of progesterone were found to produce no significant inflammation or toxicity and actually avoided the cervical and myometrial inflammatory signaling induced by a clinical progesterone gel, while nanosuspensions of TSA and progesterone led to delivery of live pups that exhibited neurotypical development [286,287]. Moreover, vaginal delivery of SKI II via SNEDDS had no toxic effect on cultured human cervical cells (HeLa) after 24 h, and a teratogenicity study revealed no effect on birth weight or the prevalence of congenital anomalies among pups born to treated dams. Collectively, these data support low toxicity of both OTR nanoliposomes and vaginal nanoformulations; however, more in-depth studies are required as technologies advance toward clinical translation.

## 14. Challenges in Therapeutic Development

Historically, the ability to study medication use in pregnancy has been limited by the ethical challenges of enrolling pregnant people in research studies and regulatory hurdles posed by the U.S. FDA edict on atosiban in 1998 (Box 1). Atosiban was not given FDA approval due to concerns regarding the number of infant deaths among pregnant people treated with atosiban compared to a placebo. That ruling generated tremendous challenges in designing studies to clinically test potential tocolytic drugs. Novel agents now require a placebo-controlled trial to show efficacy, an option that few U.S. centers would consider, and one that some U.S. IRBs resist as contradictory to the standard of care. Attempts to circumvent stringent regulations on therapeutic research have led to many studies being exported to other countries with fewer restrictions [291]. This emerging practice raises numerous ethical concerns regarding the potential exploitation of local residents. An alternative to a purely placebo-controlled trial is ‘rescue therapy’, in which researchers administer a community-standard tocolytic (e.g., magnesium sulfate) one hour after administration of the study drug. However, as seen in the Atosiban-096 trial, rescue therapy diminishes the chance of demonstrating the full benefits of the novel drug and complicates interpretation of the results [71,73]. Furthermore, because there are no FDA-approved drugs for tocolysis that are still marketed, it is not possible to conduct an active control trial for a new tocolytic therapeutic in the U.S. In a given subset of pregnant patients, the accepted standard of care to administer off-label use of tocolytics confounds the study results, as the intervention in practice may have taken place before patients meet the definition of preterm labor necessary for enrollment.

Compounding these challenges, the FDA further insisted that future approval of tocolytic drugs be contingent upon improving neonatal outcomes. Delaying PTB is not sufficient by itself for a novel drug to obtain FDA approval, and perinatal mortality occurs too infrequently to use as a measurable outcome [54]. This poses a significant logistical burden for subject enrollment. The Atosiban-096 randomized control trial ran for longer than three years and involved 35 centers in North and South America, but it only enrolled 513 women, despite broad diagnostic criteria for spontaneous preterm labor [74]. Additionally, the field of obstetrics currently lacks a laboratory measure that can be tightly linked to PTB or neonatal outcomes, making it difficult to measure the ability of a PTB therapeutic to prevent a surrogate endpoint [6].

## 15. Pathway Forward toward New PTB Therapeutics

Identifying and finding solutions to navigate the obstacles for tocolytic development is a profound challenge. The solution is most likely to come from a coordinated, unified effort between PTB researchers, regulatory agencies, tocolytic pharmaceutical companies, obstetricians and other clinicians, funding bodies—both private foundations and government agencies—academic organizations, and patient advocacy groups. A model for this kind of collaboration already exists: the International Neonatal Consortium [292]. Developed in 2015, the International Neonatal Consortium is focused on standardizing criteria and methods for regulating both neonatal medicine development and administration in collaboration with regulatory agencies [292]. This organization has already demonstrated progress in addressing the regulatory science of two other common neonatal disorders, neonatal seizures and bronchopulmonary dysplasia, for which various therapeutics have long been employed off-label [292]. A similar global and coordinated effort could be applied to advancing the regulatory science of PTB therapeutics. Without a well-defined path to regulatory approval for novel tocolytics in the U.S., workshops are needed to debate the requirements for tocolytic approval. Should neonatal morbidity and mortality remain an imperative outcome of study to earn FDA approval, alternative study designs should be considered to achieve practical and timely study execution. Global collaboration would allow for data pooling and could accelerate study completion [292]. Finally, pregnant people are crucial stakeholders in this process and should be involved in the study design.

## 16. Conclusions

PTB is a multi-faceted obstetric problem and a major cause of neonatal morbidity and mortality. At present, there are numerous tocolytics employed in the management of PTB, providing a short delay in order to transfer patients to high-level care facilities or, rarely, to effectively reduce neonatal morbidities. There is an urgent need to develop tocolytic agents that can provide a longer-lasting delay of PTB that clearly impacts neonatal and child health. These novel agents in pre-clinical development or in clinical trials include BSCI, OBE-022, IL1-R antagonist (e.g., Rytvela, Kineret), and nanoparticle delivery systems. Significant regulatory hurdles exist for bringing any new therapeutic in pregnancy to market. The few tocolytic studies that previously garnered FDA approval are no longer adequate or appropriate as a model for testing new PTB therapeutics, hindering the ability of novel therapeutic research to follow a standardized study design. A new coordinated approach between clinicians, scientists, drug development agencies, pharmaceutical companies, funding organizations, and patient advocacy groups is necessary to accelerate the development of PTB therapeutics. This collaboration will aid in developing more effective study designs and will cultivate data pooling to allow for practical and timely evaluation of novel therapeutic agents.

## Figures and Tables

**Figure 1 jcm-10-02912-f001:**
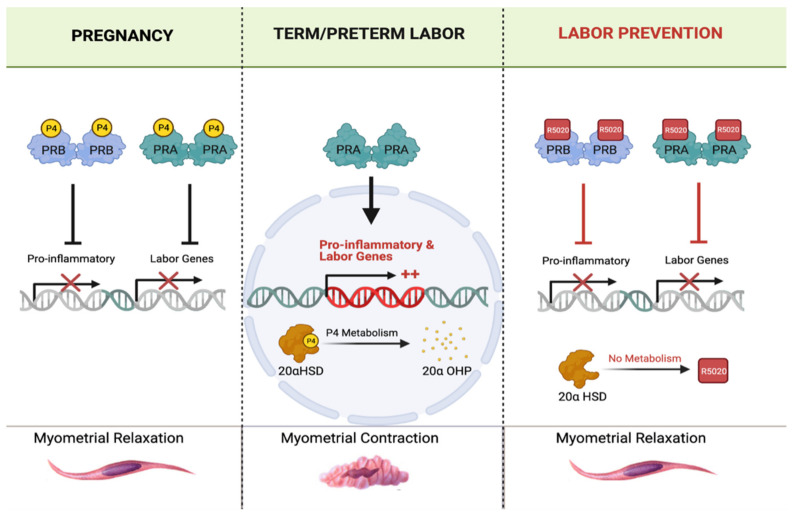
The potential mechanisms of selective P4 receptor modulator action on uterine muscle during pregnancy and term/preterm labor. (**Left panel**) During pregnancy, P4 liganding of P4 receptors (PR-A and PR-B) inhibits pro-inflammatory (cytokines and chemokines) and pro-contractile (CAPs) uterine genes, thereby maintaining myometrial relaxation. (**Middle panel**) During term and preterm labor, myometrial 20α-hydroxysteroid dehydrogenase (20α-HSD) enzyme expression and activity is upregulated, which results in local intracellular metabolism of P4 into its PR-inactive metabolite, 20α-hydroxyprogesterone (20αOHP). This leads to un-liganding of PRs (unbounding of P4). Un-liganded PR-A activates myometrial expression of pro-inflammatory and pro-contractile genes (i.e., *oxtr* and *gja1*) and induces labor contractions. (**Right panel**) Administration of SPRM compounds such as R5020 (aka: Promegestone), which is not a substrate for 20α-HSD, has higher affinity for PRs, longer half-life than P4, keeps the PRs constitutively liganded, maintains uterine quiescence, and prevents labor contractions. Note: this figure was created with Biorender.com.

**Figure 2 jcm-10-02912-f002:**
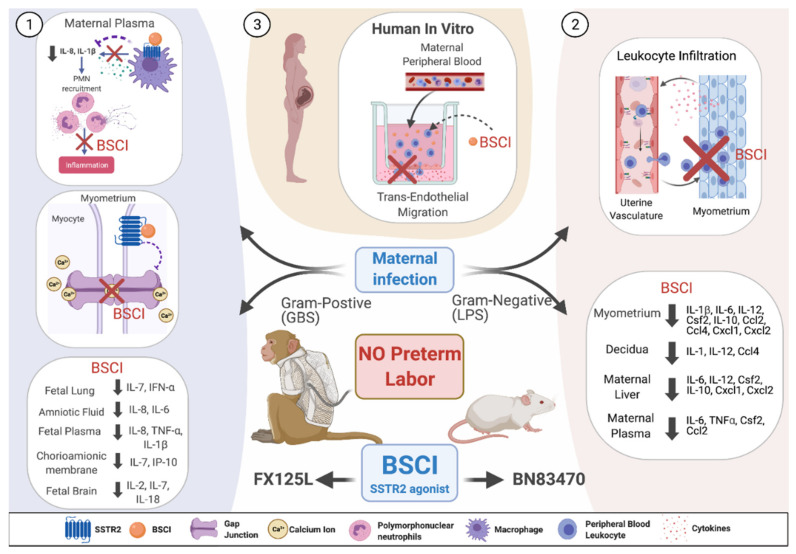
Conceptual model showing BSCI actions in vivo and in vitro as a preterm labor therapeutic. (**1**) In vivo administration of BSCI (FX125L) using a nonhuman primate model of preterm labor induced by Group B Streptococcus (GBS) led to the powerful suppression of uterine activity and a complete blockade of PTB. BSCI treatment led to reduced maternal plasma IL-8 and IL-1β inhibited myometrial gap junction protein connexin 43 mRNA levels and reduced pro-inflammatory cytokines in amniotic fluid, chorioamniotic membrane, fetal plasma, lungs, and brain compared to GBS alone [210]. (**2**) Prophylactic in vivo administration of BSCI (BN83470) decreased LPS-induced PTB in pregnant mice, significantly inhibited neutrophil infiltration in the mouse myometrium, and significantly attenuated multiple cytokine/chemokine expression in maternal tissues (myometrium, decidua, plasma, and liver) [211]. (**3**) We hypothesize that pre-treatment with BSCI (FX125L) of human primary leukocytes isolated from peripheral blood of pregnant people will also prevent the in vitro trans-endothelial migration of neutrophils towards media containing multiple cytokines secreted from the pregnant human decidua and myometrium. Note: this figure was created with Biorender.com.

**Figure 3 jcm-10-02912-f003:**
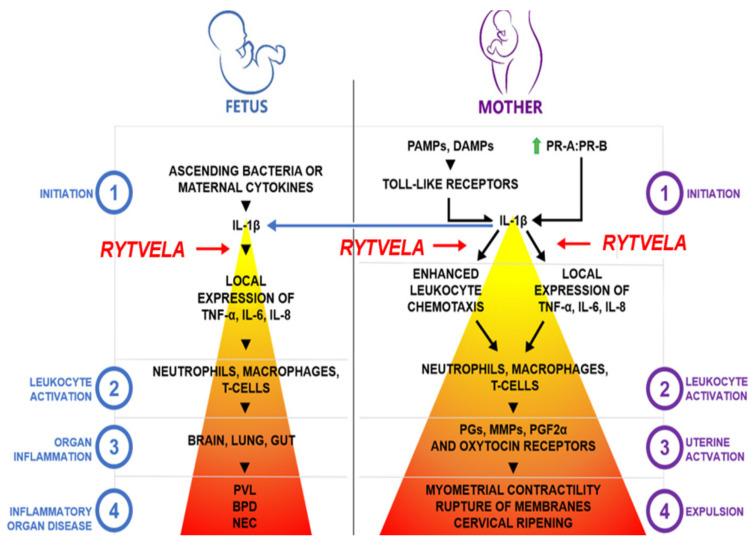
Conceptual model for the role of IL-1β in preterm labor and fetal inflammation. This model illustrates that IL-1β is the apex cytokine in the inflammatory cascade of preterm birth and fetal inflammatory injury, thereby presenting an attractive molecular target for drug discovery. PR-A/PR-B, P4 receptors A and B; PGs, prostaglandins; MMPs, matrix metalloproteinases; PGF_2α_, prostaglandin F_2α_; PLV, periventricular leukomalacia; BPD, bronchopulmonary dysplasia; and NEC, necrotizing enterocolitis. Increasing color intensity represents increasing inflammatory response. The level where Rytvela acts is identified by red arrows. Courtesy of Han Lee.

**Figure 4 jcm-10-02912-f004:**
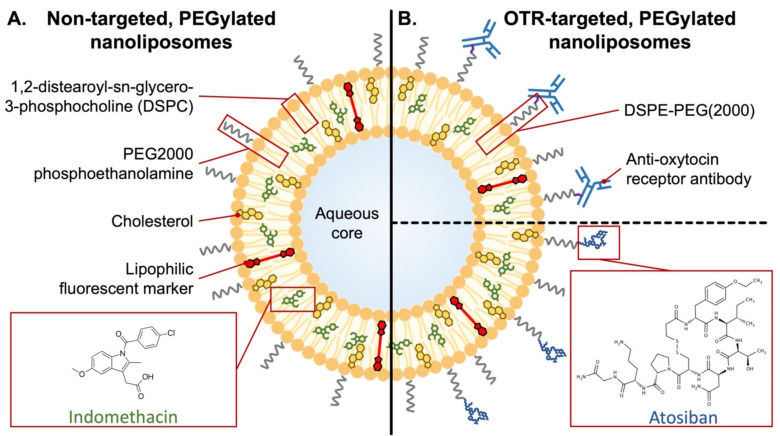
Schematic of non-targeted and uterine-targeted nanoliposomes. (**A**) The nanoliposomes are composed primarily of 1,2-distearoyl-sn-glycero-3-phosphocholine (DSPC) and cholesterol but also include PEGylated lipid (PEG2000 phosphoethanolamine; ~2% of total lipids), which produces steric hindrance that improves circulation time, but without a targeting moiety. Indomethacin and lipophilic markers partition into the lipid bilayer. (**B**) Oxytocin receptor (OTR)-targeted nanoliposomes, whereby PEGylated 1,2-distearoyl-sn-glycero-3-phosphoethanolamine (DSPE, ~2% of total lipids) is conjugated to either an OTR-binding antibody (via maleimide linkage) or peptide (Atosiban, via amine linkage).

**Table 1 jcm-10-02912-t001:** Summary of preterm birth therapeutics.

Therapeutic	Route	Mechanism of Action	Stage of Development	Side Effects	Other Notes
Terbutaline (Bricanyl, Marex)	oral, IV	beta-2 adrenergic agonist	clinical use	Tremor, shakiness	
Atosiban (Tractocile, Antocin)	IV	oxytocin and vasopressin antagonist	clinical use	Nausea, vomiting, headache	
Nifedipine (Procardia, Adalat, Afeditab)	oral	calcium channel blocker	clinical use	Headache, flushing, constipation	
Antibiotics	oral, IV	dependent on the bacterial target	clinical use	Depends on antibiotic type	Used after PPROM
Aspirin ^1^	oral	COX inhibitor	clinical use	Rash, peptic ulcers, abdominal pain, nausea	
Makena^®^ (hydroxypro-gesterone caproate) ^2^	IM	17-OH-Progesterone	clinical use	Itching, nausea, diarrhea, injection site reaction	
OBE022 (Ebopiprant)	oral	prostaglandin receptor antagonist	Phase II	Headache, constipation	
Rytvela	IV, SC	IL-1 receptor allosteric modulator	pre-clinical	None noted in mothers or offspring	None noted in mothers or offspring
Kineret^®^ (Anakinra)	SC	IL-1 receptor antagonist	clinically approved	Injection site reaction, immune suppression (increased infection risk)	Not approved (or used) to prevent preterm birth
BSCI	IV	SSTR2	pre-clinical		
Immunoliposome					

^1^ ASA. ^2^ OHPC.

**Table 2 jcm-10-02912-t002:** Novel inflammatory therapeutics and ongoing clinical trials.

Company	Study (Type)	Drug	*n*, Target Population	Design	Primary Endpoint	Findings
AMAG	MEIS P2	Makena (17-OHP)	*n* = 463 16–21 week with history of PTB	RCT (US multi-center)	Reduction of PTB (<37 week, <35 week, <32 week)	PTB RR 0.66, 0.67, 0.58; Cl 0.54–0.81, 0.48–0.93, 0.37–0.91
AMAG	PROLONG P3	Makena (17-OHP)	*n* = 1708 16–21 week with history of PTB	RCT (international multi-center)	Reduction of PTB (<35 week), neonatal morbidity	No reduction in PTB or neonatal morbidity PTB RR 0.95, Cl 0.71, 1.26
ObsEva	TERM P2	OBE-001 (oxytocin receptor antagonist)	*n* = 10 34–36 week with PRETERM LABOR	RCT	Incidence of delivery within 7 days	terminated
ObsEva	PROLONG P2	OBE-022 (PGF2_α_ receptor)	*n* = 120 28–34 week with PRETERM LABOR	RCT	Delivery < 2 days, Delivery < 7 days, delivery < 37 week, time to delivery	ongoing
GSK	P2	Retosiban (oxytocin receptor antagonist)	*n* = 64 30–35 week with PRETERM LABOR	RCT	Resolution of contractions	terminated
Ferring Pharma	P2	Barusiban (oxytocin receptor antagonist)	*n* = 163 34–36 week with PRETERM LABOR	RCT	Delivery within 48 h	No reduction in delivery within 48 h
Lipocine	P3	LPCN1107 (oral 17-OHP)	*n* = 1100	RCT 17OHP oral vs. IM	Reduction of PTB < 37 week	ongoing
Hadassah Medical Organization	P2	Indomethacin	*n* = 300 24–32 week with PRETERM LABOR	RCT Indometh vs. nifedipine	Time to delivery, GA at delivery	proposed
University of Hong Kong	P3	Oral dydrogesterone	*n* = 1714 <14 week	RCT	Rate of PTB < 37 week	ongoing
NICHD	ASPIRIN	Oral Aspirin	*n* = 11,976 6–14 week	RCT (international multi-center)	Rate of PTB < 37 week	PTB RR 0.89, Cl 0.81–0.98

## Data Availability

Not applicable.

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
