# Peer review of "Landscape of Preterm Birth Therapeutics and a Path Forward"

_jcm, 2021, doi:10.3390/jcm10132912_

Round 1

Reviewer 1 Report

This is an extremely well written and thorough review of current and emerging agents to prevent preterm birth.  It provides just the right amount of detail about mechanisms for the general reader, and is well referenced for the reader who wants more detail.  The review ends with a practical discussion of some of the study challenges and regulatory hurdles involved in medical research and therapeutics for pregnant persons, and a call-to-action.

The flow is logical, the figures beautiful, and I am certain that the few formatting errors in tables/boxes will be corrected in a final version.  I have only 2 small, picky comments:

1. Please change pregnant “women” to pregnant “persons” on line 893. 

2. Consider making a minor modification to the Pathophysiology of PTB section to make it even clearer that the parturition cascade involves activation of and communication between all of the reproductive tissues (cervix and membranes as well as myometrium).  This paper is heavy on myometrial activation (for the obvious reason), and only mentions cervical dilation (line 115) as evidence of cervical involvement in the parturition cascade.  Another sentence or two about the cervix (it softens and shortens as well as dilates) would clarify both the argument for progesterone in section 9, where its use in patients with a sonographic short cervix is discussed, and enhance the discussions of the emerging agents which target upstream initiators of the pro-inflammatory cascade which targets all of the reproductive tissues.

Author Response

Dear Reviewer 1,

Thank you for your thoughtful feedback on our manuscript. Below, we have responded to your comments and made revisions to our manuscript. We hope these changes will be acceptable for publication in JCM.

Reviewer 1 Comment: This is an extremely well written and thorough review of current and emerging agents to prevent preterm birth.  It provides just the right amount of detail about mechanisms for the general reader, and is well referenced for the reader who wants more detail.  The review ends with a practical discussion of some of the study challenges and regulatory hurdles involved in medical research and therapeutics for pregnant persons, and a call-to-action.

The flow is logical, the figures beautiful, and I am certain that the few formatting errors in tables/boxes will be corrected in a final version.  

Authors’ Response: We appreciate these kind sentiments and thank the Reviewer for their generous comments on our manuscript. Formatting errors (ex. the missing beta symbol in IL-1β) have been identified and will be addressed in the final version.

Reviewer 1 Comment 1: Please change pregnant “women” to pregnant “persons” on line 893. 

Authors’ Response: This is an insightful suggestion, reflecting a push for more progressive gender-inclusive language. We have identified numerous instances where pregnant “women” could be changed to pregnant “people” and have made those changes as appropriate. We thank the Reviewer for noting this.

Reviewer 1 Comment 2:  Consider making a minor modification to the Pathophysiology of PTB section to make it even clearer that the parturition cascade involves activation of and communication between all of the reproductive tissues (cervix and membranes as well as myometrium).  This paper is heavy on myometrial activation (for the obvious reason), and only mentions cervical dilation (line 115) as evidence of cervical involvement in the parturition cascade.  Another sentence or two about the cervix (it softens and shortens as well as dilates) would clarify both the argument for progesterone in section 9, where its use in patients with a sonographic short cervix is discussed, and enhance the discussions of the emerging agents which target upstream initiators of the pro-inflammatory cascade which targets all of the reproductive tissues.

Authors’ Response:  We agree that additional text on the cervical physiology is important for rounding out the multi-faceted pathways to PTB while simultaneously supporting subsequent sections in the manuscript. We have revised the text accordingly.

Reviewer 2 Report

This article aims to review the current landscape of PTB therapeutics, including novel agents in pre-clinical development or clinical trials, and to reflect on challenges of therapeutic development and ways forward.

I would like to congratulate the authors on a well-written and extensively documented piece of work, which probably required a massive amount of work. My comments and suggestions are as follows:

  1. The term “preterm birth therapeutics” is a bit vague, in the sense that it can refer to therapeutics aiming at preventing preterm birth or preventing the neonatal consequences of PTB.
  2. I would recommend that the scope of this review is a little more focused There is a clear imbalance between sections on tocolytics (90% of the article) and other therapeutics, such as antibiotics or aspirin. Most of the studies and conclusions presented here are related to women with SPL and intact membranes. It would be interesting to either “only” address tocolytics in women with SPL, or to provide more information on women with PPROM.
  3. In the introduction, the authors might consider adding a few sentences about the etiology of PTB and the need to adapt therapeutics to the etiology. Most of the treatments presented here are not indicated in women with other causes of PTB than SPL and PPROM (such as FGR or hypertensive disorders), which justifies to focus the review. They could state clearly that the objective of tocolysis administration is to improve neonatal outcomes (which has hardly been demonstrated) and not to prolong pregnancy. In women with PPROM, when antibiotics are administered, there is less and less evidence that tocolytics are associated with a prolongation of the latency period or benefits to the newborn (10.1002/14651858.CD007062.pub3, http://dx.doi.org/10.1016/j.ajog.2017.04.015, https://doi.org/10.1038/s41598-020-65201-y).
  4. Table 1: add indications for each therapeutic and abbreviations in footnote. Table 2: add 95%CI in the findings column where appropriate + complete this column for Lipocine.
  1. Section 3: It would be great to provide a clear definition of tocolytic efficacy, and to indicate the reference group when comparisons are reported (e.g., “betamimetics have not been shown to reduce…” -> compared to placebo?).
  2. Section 7: when RRs are reported, I would recommend presenting the number of studies and the number of women analysed. I’m not sure that the few sentences on fetal neuroprotection are necessary. A short summary at the end of the section is lacking.
  3. The authors, if they wish to keep this section, could mention the risks associated with antibiotic administration in the absence of a clear indication (microbiota, resistances, etc). In women with PPROM, there is a clear gap in knowledge regarding the antibiotic agents, regimens and doses to be used.
  4. In section 11, some sentences are very redundant (on the PROLONG trial and OBE-022). Links to clinicaltrials.org could be provided.
  5. Figure 3: there is an issue with the β in the footnote, same in section 12/Rytvela.
  6. Section 14 could include a few hints on non US countries.
  7. Conclusion: I would state again that most tocolytics have no impact on neonatal outcomes.
  8. Several authors have conflicts of interest with the pharmaceutical industry. I would provide more information on their role in performing the review.

Author Response

Dear Reviewer 2,

Thank you for your thoughtful feedback on our manuscript. Below, we have responded to the Reviewer comments and hope these changes will be acceptable for publication in JCM.

Reviewer 2 Comment:  This article aims to review the current landscape of PTB therapeutics, including novel agents in pre-clinical development or clinical trials, and to reflect on challenges of therapeutic development and ways forward. I would like to congratulate the authors on a well-written and extensively documented piece of work, which probably required a massive amount of work.

Authors’ Response: We thank the Reviewer for these thoughtful comments on our manuscript.

Reviewer 2 Comment 1: The term “preterm birth therapeutics” is a bit vague, in the sense that it can refer to therapeutics aiming at preventing preterm birth or preventing the neonatal consequences of PTB.

Authors’ Response: We appreciate the challenges in naming this class of therapeutics. Nevertheless, we have not identified a more clear or succinct term to describe this, and we have included a sentence in our introduction to discuss ‘preterm birth therapeutics’ as agents which aid in delaying PTB and may contribute to reducing maternal and neonatal consequences of PTB. Moreover, lines 57-59 highlight the scarcity of research to suggest PTB therapeutics can aid in reducing neonatal complications. We have included an additional comment affirming this point in the conclusion. We thank the Reviewer for highlighting this potential area of confusion.

Reviewer 2 Comment 2: I would recommend that the scope of this review is a little more focused. There is a clear imbalance between sections on tocolytics (90% of the article) and other therapeutics, such as antibiotics or aspirin. Most of the studies and conclusions presented here are related to women with SPL and intact membranes. It would be interesting to either “only” address tocolytics in women with SPL, or to provide more information on women with PPROM.

Authors’ Response: We recognize this review of preterm birth therapeutics is extensive, covering mostly tocolytics for women with SPL and intact membranes. However, we prefer to keep the sections on antibiotics and aspirin. We have highlighted that our main focus is on tocolytics for SPL and intact membranes now in the introduction.

Reviewer 2 Comment 3: In the introduction, the authors might consider adding a few sentences about the etiology of PTB and the need to adapt therapeutics to the etiology. Most of the treatments presented here are not indicated in women with other causes of PTB than SPL and PPROM (such as FGR or hypertensive disorders), which justifies to focus the review. They could state clearly that the objective of tocolysis administration is to improve neonatal outcomes (which has hardly been demonstrated) and not to prolong pregnancy. In women with PPROM, when antibiotics are administered, there is less and less evidence that tocolytics are associated with a prolongation of the latency period or benefits to the newborn (10.1002/14651858.CD007062.pub3, http://dx.doi.org/10.1016/j.ajog.2017.04.015, https://doi.org/10.1038/s41598-020-65201-y).

Authors’ Response: We agree. Additional text has now been included in the introduction to address this point. We also acknowledge that the literature is built around correlating therapeutic use with a delay in PTB, not necessarily neonatal outcomes, which is why we describe this. However, the goal of PTB therapeutic use is clearly to improve neonatal outcomes. Remarks on antibiotic use for women with PPROM have been addressed in the context of another comment left by the Reviewer below (Comment 7).

Reviewer 2 Comment 4: Table 1: add indications for each therapeutic and abbreviations in footnote. Table 2: add 95%CI in the findings column where appropriate + complete this column for Lipocine.

Authors’ Response: Abbreviations and indications for these therapeutics are indicated now in the footnote. For Table 2, 95% confidence intervals have been included when reported by the authors. The column for Lipocine has been completed.

Reviewer 2 Comment 5: Section 3: It would be great to provide a clear definition of tocolytic efficacy, and to indicate the reference group when comparisons are reported (e.g., “betamimetics have not been shown to reduce…” -> compared to placebo?).

Authors’ Response: We thank the Reviewer for highlighting the need to add reference groups when discussing efficacy and side-effects. Updates have been made to reflect this suggestion.

Reviewer 2 Comment 6: Section 7: when RRs are reported, I would recommend presenting the number of studies and the number of women analyzed. I’m not sure that the few sentences on fetal neuroprotection are necessary. A short summary at the end of the section is lacking.

Authors’ Response: We agree and have now included this information for both the study where relative risk was mentioned directly as well as for the other meta-analysis referenced frequently throughout the section. We have opted to keep the section on fetal neuroprotection included in the manuscript. A short summary at the end of the section has been added.

Reviewer 2 Comment 7: The authors, if they wish to keep this section, could mention the risks associated with antibiotic administration in the absence of a clear indication (microbiota, resistances, etc). In women with PPROM, there is a clear gap in knowledge regarding the antibiotic agents, regimens and doses to be used.

Authors’ Response: Thank you for this suggestion and we have now addressed the associated risks of antibiotic administration in patients without clear signs of infection as well as the controversial use of tocolytics in delaying PTB among patients with PPROM. Section 8 on antibiotics and PPROM has been edited to encompass new language that addresses the points raised by the Reviewer.

Reviewer 2 Comment 8: In section 11, some sentences are very redundant (on the PROLONG trial and OBE-022). Links to clinicaltrials.org could be provided.

Authors’ Response: We thank the Reviewer for identifying repetitive sentences in this section. This section has been edited to remove redundancy. Links to clinicaltrials.org are challenging to cite properly through the EndNote program.

Reviewer 2 Comment 9: Figure 3: there is an issue with the β in the footnote, same in section 12/Rytvela.

Authors’ Response: As fonts changed from Arial to Palatino in the Editorial process post-manuscript submission, the beta symbol was lost. This issue has been identified in numerous instances throughout the manuscript and will be address in the final version. We thank the Reviewer for noting this.

Reviewer 2 Comment 10: Section 14 could include a few hints on non US countries.

Authors’ Response: As novel therapeutic development studies in the United States face significant scrutiny from the FDA, a growing trend of exporting studies to sub-Saharan Africa has occurred recently. These attempts to circumvent stringent FDA regulations by taking therapeutic trials abroad to other nations with fewer regulations raise significant ethical concerns. Inadequate healthcare infrastructure in certain countries has made it easier to exploit local residents, forcing them to participate in risk-laden drug studies in order to qualify for their medical treatments. The unknown safety profile and side effects of novel therapeutics are real health risks to real people; as U.S. pharmaceutical companies export that risk abroad, researchers must be extremely cautious in recruiting study participants from populations that may be afflicted by extreme poverty or inadequate access to health care. In our manuscript, we have highlighted this sentiment with two additional sentences.

Reviewer 2 Comment 11: Conclusion: I would state again that most tocolytics have no impact on neonatal outcomes.

Authors’ Response: We now highlight this item in the introduction and conclusion.

Reviewer 2 Comment 12: Several authors have conflicts of interest with the pharmaceutical industry. I would provide more information on their role in performing the review.

Authors’ Response: Updates have been included to the Conflicts of Interest section.